# DeepSphere: a graph-based spherical CNN

**Michaël Defferrard, Martino Milani & Frédérick Gusset**
École Polytechnique Fédérale de Lausanne (EPFL), Switzerland
{michael.defferrard,martino.milani,frederick.gusset}@epfl.ch

**Nathanaël Perraudin**
Swiss Data Science Center (SDSC), Switzerland
nathanael.perraudin@sdsc.ethz.ch

## Abstract

Designing a convolution for a spherical neural network requires a delicate trade-off between efficiency and rotation equivariance. DeepSphere, a method based on a graph representation of the sampled sphere, strikes a controllable balance between these two desiderata. This contribution is twofold. First, we study both theoretically and empirically how equivariance is affected by the underlying graph with respect to the number of vertices and neighbors. Second, we evaluate DeepSphere on relevant problems. Experiments show state-of-the-art performance and demonstrates the efficiency and flexibility of this formulation. Perhaps surprisingly, comparison with previous work suggests that anisotropic filters might be an unnecessary price to pay. Our code is available at https://github.com/deepsphere.

## 1 Introduction

Spherical data is found in many applications (figure 1). Planetary data (such as meteorological or geological measurements) and brain activity are example of intrinsically spherical data. The observation of the universe, LIDAR scans, and the digitalization of 3D objects are examples of projections due to observation. Labels or variables are often to be inferred from them. Examples are the inference of cosmological parameters from the distribution of mass in the universe (Perraudin et al., 2019), the segmentation of omnidirectional images (Khasanova & Frossard, 2017), and the segmentation of cyclones from Earth observation (Mudigonda et al., 2017).

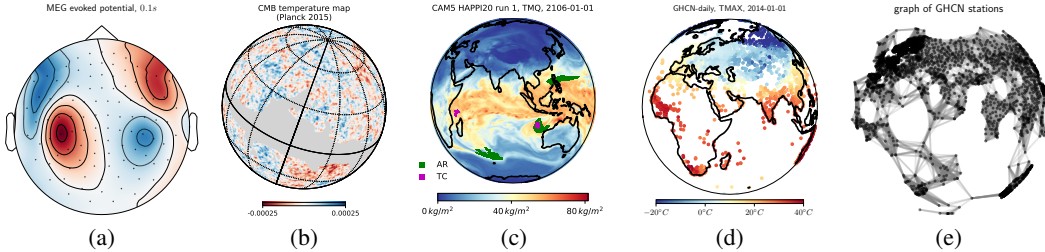

Figure 1: Examples of spherical data: (a) brain activity recorded through magnetoencephalography (MEG),[1] (b) the cosmic microwave background (CMB) temperature from Planck Collaboration (2016), (c) hourly precipitation from a climate simulation (Jiang et al., 2019), (d) daily maximum temperature from the Global Historical Climatology Network (GHCN).[2] A rigid full-sphere sampling is not ideal: brain activity is only measured on the scalp, the Milky Way's galactic plane masks observations, climate scientists desire a variable resolution, and the position of weather stations is arbitrary and changes over time. (e) Graphs can faithfully and efficiently represent sampled spherical data by placing vertices where it matters.

As neural networks (NNs) have proved to be great tools for inference, variants have been developed to handle spherical data. Exploiting the locally Euclidean property of the sphere, early attempts used standard 2D convolutions on a grid sampling of the sphere (Boomsma & Frellsen, 2017; Su & Grauman, 2017; Coors et al., 2018). While simple and efficient, those convolutions are not equivariant to rotations. On the other side of this tradeoff, Cohen et al. (2018) and Esteves et al. (2018) proposed to perform proper spherical convolutions through the spherical harmonic transform. While equivariant to rotations, those convolutions are expensive (section 2).

As a lack of equivariance can penalize performance (section 4.2) and expensive convolutions prohibit their application to some real-world problems, methods standing between these two extremes are desired. Cohen et al. (2019) proposed to reduce costs by limiting the size of the representation of the symmetry group by projecting the data from the sphere to the icosahedron. The distortions introduced by this projection might however hinder performance (section 4.3).

Another approach is to represent the sampled sphere as a graph connecting pixels according to the distance between them (Bruna et al., 2013; Khasanova & Frossard, 2017; Perraudin et al., 2019). While Laplacian-based graph convolutions are more efficient than spherical convolutions, they are not exactly equivariant (Defferrard et al., 2019). In this work, we argue that graph-based spherical CNNs strike an interesting balance, with a controllable tradeoff between cost and equivariance (which is linked to performance). Experiments on multiple problems of practical interest show the competitiveness and flexibility of this approach.

## 2 METHOD

DeepSphere leverages graph convolutions to achieve the following properties: (i) computational efficiency, (ii) sampling flexibility, and (iii) rotation equivariance (section 3). The main idea is to model the sampled sphere as a graph of connected pixels: the length of the shortest path between two pixels is an approximation of the geodesic distance between them. We use the graph CNN formulation introduced in (Defferrard et al., 2016) and a pooling strategy that exploits hierarchical samplings of the sphere.

**Sampling.** A sampling scheme $\mathcal{V} = \{x_i \in \mathbb{S}^2\}_{i=1}^n$ is defined to be the discrete subset of the sphere containing the $n$ points where the values of the signals that we want to analyse are known. For a given continuous signal $f$, we represent such values in a vector $\boldsymbol{f} \in \mathbb{R}^n$. As there is no analogue of uniform sampling on the sphere, many samplings have been proposed with different tradeoffs. In this work, depending on the considered application, we will use the equiangular (Driscoll & Healy, 1994), HEALPix (Gorski et al., 2005), and icosahedral (Baumgardner & Frederickson, 1985) samplings.

**Graph.** From $\mathcal{V}$, we construct a weighted undirected graph $\mathcal{G} = (\mathcal{V}, w)$, where the elements of $\mathcal{V}$ are the vertices and the weight $w_{ij} = w_{ji}$ is a similarity measure between vertices $x_i$ and $x_j$. The combinatorial graph Laplacian $\boldsymbol{L} \in \mathbb{R}^{n \times n}$ is defined as $\boldsymbol{L} = \boldsymbol{D} - \boldsymbol{A}$, where $\boldsymbol{A} = (w_{ij})$ is the weighted adjacency matrix, $\boldsymbol{D} = (d_{ii})$ is the diagonal degree matrix, and $d_{ii} = \sum_j w_{ij}$ is the weighted degree of vertex $x_i$. Given a sampling $\mathcal{V}$, usually fixed by the application or the available measurements, the freedom in constructing $\mathcal{G}$ is in setting $w$. Section 3 shows how to set $w$ to minimize the equivariance error.

**Convolution.** On Euclidean domains, convolutions are efficiently implemented by sliding a window in the signal domain. On the sphere however, there is no straightforward way to implement a convolution in the signal domain due to non-uniform samplings. Convolutions are most often performed in the spectral domain through a spherical harmonic transform (SHT). That is the approach taken by Cohen et al. (2018) and Esteves et al. (2018), which has a computational cost of $\mathcal{O}(n^{3/2})$ on isolatitude samplings (such as the HEALPix and equiangular samplings) and $\mathcal{O}(n^2)$ in general.

---

[1] https://martinos.org/mne/stable/auto_tutorials/plot_visualize_evoked.html
[2] https://www.ncdc.noaa.gov/ghcn-daily-description

On the other hand, following Defferrard et al. (2016), graph convolutions can be defined as

$$h(\boldsymbol{L})\boldsymbol{f} = \left( \sum_{i=0}^{P} \alpha_i \boldsymbol{L}^i \right) \boldsymbol{f}, \tag{1}$$

where $P$ is the polynomial order (which corresponds to the filter's size) and $\alpha_i$ are the coefficients to be optimized during training.[3] Those convolutions are used by Khasanova & Frossard (2017) and Perraudin et al. (2019) and cost $\mathcal{O}(n)$ operations through a recursive application of $\boldsymbol{L}$.[4]

**Pooling.** Down- and up-sampling is natural for hierarchical samplings,[5] where each subdivision divides a pixel in (an equal number of) child sub-pixels. To pool (down-sample), the data supported on the sub-pixels is summarized by a permutation invariant function such as the maximum or the average. To unpool (up-sample), the data supported on a pixel is copied to all its sub-pixels.

**Architecture.** All our NNs are fully convolutional, and employ a global average pooling (GAP) for rotation invariant tasks. Graph convolutional layers are always followed by batch normalization and ReLU activation, except in the last layer. Note that batch normalization and activation act on the elements of $\boldsymbol{f}$ independently, and hence don't depend on the domain of $f$.

## 3 GRAPH CONVOLUTION AND EQUIVARIANCE

While the graph framework offers great flexibility, its ability to faithfully represent the underlying sphere — for graph convolutions to be rotation equivariant — highly depends on the sampling locations and the graph construction.

### 3.1 PROBLEM FORMULATION

A continuous function $f : \mathcal{C}(\mathbb{S}^2) \supset F_{\mathcal{V}} \to \mathbb{R}$ is sampled as $T_{\mathcal{V}}(f) = \boldsymbol{f}$ by the sampling operator $T_{\mathcal{V}} : C(\mathbb{S}^2) \supset F_{\mathcal{V}} \to \mathbb{R}^n$ defined as $\boldsymbol{f} : f_i = f(x_i)$. We require $F_{\mathcal{V}}$ to be a suitable subspace of continuous functions such that $T_{\mathcal{V}}$ is invertible, i.e., the function $f \in F_{\mathcal{V}}$ can be unambiguously reconstructed from its sampled values $\boldsymbol{f}$. The existence of such a subspace depends on the sampling $\mathcal{V}$ and its characterization is a common problem in signal processing (Driscoll & Healy, 1994). For most samplings, it is not known if $F_{\mathcal{V}}$ exists and hence if $T_{\mathcal{V}}$ is invertible. A special case is the equiangular sampling where a sampling theorem holds, and thus a closed-form of $T_{\mathcal{V}}^{-1}$ is known. For samplings where no such sampling formula is available, we leverage the discrete SHT to reconstruct $f$ from $\boldsymbol{f} = T_{\mathcal{V}}f$, thus approximating $T_{\mathcal{V}}^{-1}$. For all theoretical considerations, we assume that $F_{\mathcal{V}}$ exists and $f \in F_{\mathcal{V}}$.

By definition, the (spherical) graph convolution is rotation equivariant if and only if it commutes with the rotation operator defined as $R(g), g \in SO(3)$: $R(g)f(x) = f\left(g^{-1}x\right)$. In the context of this work, graph convolution is performed by recursive applications of the graph Laplacian (1). Hence, if $R(g)$ commutes with $\boldsymbol{L}$, then, by recursion, it will also commute with the convolution $h(\boldsymbol{L})$. As a result, $h(\boldsymbol{L})$ is rotation equivariant if and only if

$$\boldsymbol{R}_{\mathcal{V}}(g)\boldsymbol{L}\boldsymbol{f} = \boldsymbol{L}\boldsymbol{R}_{\mathcal{V}}(g)\boldsymbol{f}, \qquad \forall f \in F_{\mathcal{V}} \text{ and } \forall g \in SO(3),$$

where $\boldsymbol{R}_{\mathcal{V}}(g) = T_{\mathcal{V}}R(g)T_{\mathcal{V}}^{-1}$. For an empirical evaluation of equivariance, we define the *normalized equivariance error* for a signal $\boldsymbol{f}$ and a rotation $g$ as

$$E_{\boldsymbol{L}}(\boldsymbol{f}, g) = \left( \frac{\|\boldsymbol{R}_{\mathcal{V}}(g)\boldsymbol{L}\boldsymbol{f} - \boldsymbol{L}\boldsymbol{R}_{\mathcal{V}}(g)\boldsymbol{f}\|}{\|\boldsymbol{L}\boldsymbol{f}\|} \right)^2. \tag{2}$$

More generally for a class of signals $f \in C \subset F_{\mathcal{V}}$, the *mean equivariance error* defined as

$$\overline{E}_{\boldsymbol{L},C} = \mathbb{E}_{\boldsymbol{f} \in C, g \in SO(3)} \, E_{\boldsymbol{L}}(\boldsymbol{f}, g) \tag{3}$$

represents the overall equivariance error. The expected value is obtained by averaging over a finite number of random functions and random rotations.

---

[3]In practice, training with Chebyshev polynomials (instead of monomials) is slightly more stable. We believe it to be due to their orthogonality and uniformity.

[4]As long as the graph is sparsified such that the number of edges, i.e., the number of non-zeros in $\boldsymbol{A}$, is proportional to the number of vertices $n$. This can always be done as most weights are very small.

[5]The equiangular, HEALPix, and icosahedral samplings are of this kind.

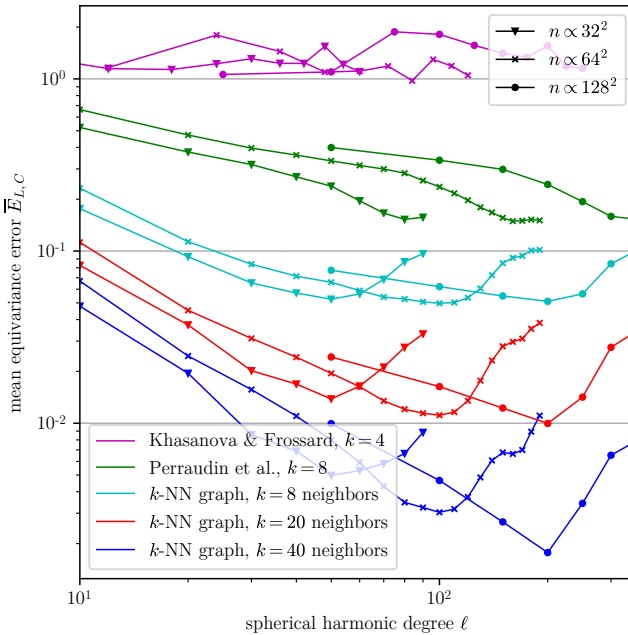

Figure 2: Mean equivariance error (3). There is a clear tradeoff between equivariance and computational cost, governed by the number of vertices $n$ and edges $kn$.

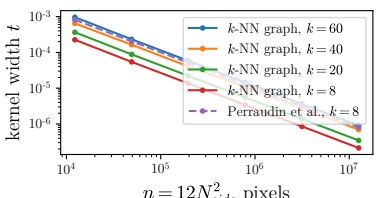

Figure 3: Kernel widths.

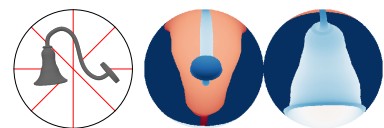

Figure 4: 3D object represented as a spherical depth map.

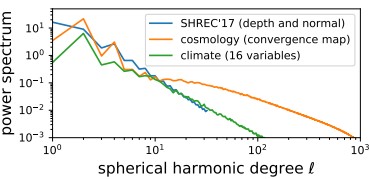

Figure 5: Power spectral densities.

## 3.2 FINDING THE OPTIMAL WEIGHTING SCHEME

Considering the equiangular sampling and graphs where each vertex is connected to 4 neighbors (north, south, east, west), Khasanova & Frossard (2017) designed a weighting scheme to minimize (3) for longitudinal and latitudinal rotations[6]. Their solution gives weights inversely proportional to Euclidean distances:

$$w_{ij} = \frac{1}{\|x_i - x_j\|}. \tag{4}$$

While the resulting convolution is not equivariant to the whole of $SO(3)$ (figure 2), it is enough for omnidirectional imaging because, as gravity consistently orients the sphere, objects only rotate longitudinally or latitudinally.

To achieve equivariance to all rotations, we take inspiration from Belkin & Niyogi (2008). They prove that for a *random uniform sampling*, the graph Laplacian $L$ built from weights

$$w_{ij} = e^{-\frac{1}{4t}\|x_i - x_j\|^2} \tag{5}$$

converges to the Laplace-Beltrami operator $\Delta_{\mathbb{S}^2}$ as the number of samples grows to infinity. This result is a good starting point as $\Delta_{\mathbb{S}^2}$ commutes with rotation, i.e., $\Delta_{\mathbb{S}^2} R(g) = R(g)\Delta_{\mathbb{S}^2}$. While the weighting scheme is full (i.e., every vertex is connected to every other vertex), most weights are small due to the exponential. We hence make an approximation to limit the cost of the convolution (1) by only considering the $k$ nearest neighbors ($k$-NN) of each vertex. Given $k$, the optimal kernel width $t$ is found by searching for the minimizer of (3). Figure 3 shows the optimal kernel widths found for various resolutions of the HEALPix sampling. As predicted by the theory, $t_n \propto n^\beta, \beta \in \mathbb{R}$. Importantly however, the optimal $t$ also depends on the number of neighbors $k$.

Considering the HEALPix sampling, Perraudin et al. (2019) connected each vertex to their 8 adjacent vertices in the tiling of the sphere, computed the weights with (5), and heuristically set $t$ to half the average squared Euclidean distance between connected vertices. This heuristic however overestimates $t$ (figure 3) and leads to an increased equivariance error (figure 2).

---

[6]Equivariance to longitudinal rotation is essentially given by the equiangular sampling.

### 3.3 ANALYSIS OF THE PROPOSED WEIGHTING SCHEME

We analyze the proposed weighting scheme both theoretically and empirically.

**Theoretical convergence.** We extend the work of (Belkin & Niyogi, 2008) to a sufficiently regular, deterministic sampling. Following their setting, we work with the *extended graph Laplacian* operator as the linear operator $L_n^t : L^2(\mathbb{S}^2) \to L^2(\mathbb{S}^2)$ such that

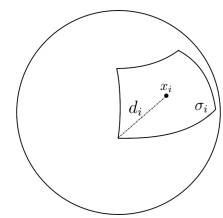

$$L_n^t f(y) := \frac{1}{n} \sum_{i=1}^n e^{-\frac{\|x_i - y\|^2}{4t}} \left( f(y) - f(x_i) \right). \qquad (6)$$

This operator extends the graph Laplacian with the weighting scheme (5) to each point of the sphere (i.e., $\boldsymbol{L}_n^t \boldsymbol{f} = T_{\mathcal{V}} L_n^t f$). As the radius of the kernel $t$ will be adapted to the number of samples, we scale the operator as

Figure 6: Patch.

$\hat{L}_n^t := |\mathbb{S}^2|(4\pi t^2)^{-1}L_n^t$. Given a sampling $\mathcal{V}$, we define $\sigma_i$ to be the patch of the surface of the sphere corresponding to $x_i$, $A_i$ its corresponding area, and $d_i$ the largest distance between the center $x_i$ and any point on the surface $\sigma_i$. Define $d^{(n)} := \max_{i=1,\dots,n} d_i$ and $A^{(n)} := \max_{i=1,\dots,n} A_i$.

**Theorem 3.1.** *For a sampling $\mathcal{V}$ of the sphere that is equi-area and such that $d^{(n)} \leq \frac{C}{n^\alpha}$, $\alpha \in (0, 1/2]$, for all $f : \mathbb{S}^2 \to \mathbb{R}$ Lipschitz with respect to the Euclidean distance in $\mathbb{R}^3$, for all $y \in \mathbb{S}^2$, there exists a sequence $t_n = n^\beta$, $\beta \in \mathbb{R}$ such that*

$$\lim_{n \to \infty} \hat{L}_n^{t_n} f(y) = \Delta_{\mathbb{S}^2} f(y).$$

This is a major step towards equivariance, as the Laplace-Beltrami operator commutes with rotation. Based on this property, we show the equivariance of the scaled extended graph Laplacian.

**Theorem 3.2.** *Under the hypothesis of theorem 3.1, the scaled graph Laplacian commutes with any rotation, in the limit of infinite sampling, i.e.,*

$$\forall y \in \mathbb{S}^2 \quad \left| R(g)\hat{L}_n^{t_n} f(y) - \hat{L}_n^{t_n} R(g)f(y) \right| \xrightarrow{n \to \infty} 0.$$

From this theorem, it follows that the discrete graph Laplacian will be equivariant in the limit of $n \to \infty$ as by construction $\boldsymbol{L}_n^t \boldsymbol{f} = T_{\mathcal{V}} L_n^t f$ and as the scaling does not affect the equivariance property of $\boldsymbol{L}_n^t$.

Importantly, the proof of Theorem 3.1 (in Appendix A) inspires our construction of the graph Laplacian. In particular, it tells us that $t$ should scale as $n^\beta$, which has been empirically verified (figure 3). Nevertheless, it is important to keep in mind the limits of Theorem 3.1 and 3.2. Both theorems present asymptotic results, but in practice we will always work with finite samplings. Furthermore, since this method is based on the capability of the eigenvectors of the graph Laplacian to approximate the spherical harmonics, a stronger type of convergence of the graph Laplacian would be preferable, i.e., spectral convergence (that is proved for a full graph in the case of random sampling for a class of Lipschitz functions in (Belkin & Niyogi, 2007)). Finally, while we do not have a formal proof for it, we strongly believe that the HEALPix sampling does satisfy the hypothesis $d^{(n)} \leq \frac{C}{n^\alpha}$, $\alpha \in (0, 1/2]$, with $\alpha$ very close or equal to $1/2$. The empirical results discussed in the next paragraph also points in this direction. This is further discussed in Appendix A.

**Empirical convergence.** Figure 2 shows the equivariance error (3) for different parameter sets of DeepSphere for the HEALPix sampling as well as for the graph construction of Khasanova & Frossard (2017) for the equiangular sampling. The error is estimated as a function of the sampling resolution and signal frequency. The resolution is controlled by the number of pixels $n = 12N_{side}^2$ for HEALPix and $n = 4b^2$ for the equiangular sampling. The frequency is controlled by setting the set $C$ to functions $f$ made of spherical harmonics of a single degree $\ell$. To allow for an almost perfect implementation (up to numerical errors) of the operator $\boldsymbol{R}_{\mathcal{V}}$, the degree $\ell$ was chosen in the range $(0, 3N_{side} - 1)$ for HEALPix and $(0, b)$ for the equiangular sampling (Gorski et al., 1999). Using these parameters, the measured error is mostly due to imperfections in the empirical approximation of the Laplace-Beltrami operator and not to the sampling.

| | performance | | size | speed | |
|---|---|---|---|---|---|
| | F1 | mAP | params | inference | training |
| Cohen et al. (2018) ($b = 128$) | - | 67.6 | 1400 k | 38.0 ms | 50 h |
| Cohen et al. (2018) (simplified,[9]$b = 64$) | 78.9 | 66.5 | 400 k | 12.0 ms | 32 h |
| Esteves et al. (2018) ($b = 64$) | 79.4 | 68.5 | 500 k | 9.8 ms | 3 h |
| DeepSphere (equiangular, $b = 64$) | 79.4 | 66.5 | 190 k | 0.9 ms | 50 m |
| DeepSphere (HEALPix, $N_{side} = 32$) | 80.7 | 68.6 | 190 k | 0.9 ms | 50 m |

Table 1: Results on SHREC'17 (3D shapes). DeepSphere achieves similar performance at a much lower cost, suggesting that anisotropic filters are an unnecessary price to pay.

Figure 2 shows that the weighting scheme (4) from (Khasanova & Frossard, 2017) does indeed not lead to a convolution that is equivariant to all rotations $g \in SO(3)$.[7] For $k = 8$ neighbors, selecting the optimal kernel width $t$ improves on (Perraudin et al., 2019) at no cost, highlighting the importance of this parameter. Increasing the resolution decreases the equivariance error in the high frequencies, an effect most probably due to the sampling. Most importantly, the equivariance error decreases when connecting more neighbors. Hence, the number of neighbors $k$ gives us a precise control of the tradeoff between cost and equivariance.

## 4 EXPERIMENTS

### 4.1 3D OBJECTS RECOGNITION

The recognition of 3D shapes is a rotation invariant task: rotating an object doesn't change its nature. While 3D shapes are usually represented as meshes or point clouds, representing them as spherical maps (figure 4) naturally allows a rotation invariant treatment.

The SHREC'17 shape retrieval contest (Savva et al., 2017) contains 51,300 randomly oriented 3D models from ShapeNet (Chang et al., 2015), to be classified in 55 categories (tables, lamps, airplanes, etc.). As in (Cohen et al., 2018), objects are represented by 6 spherical maps. At each pixel, a ray is traced towards the center of the sphere. The distance from the sphere to the object forms a depth map. The cos and sin of the surface angle forms two normal maps. The same is done for the object's convex hull.[8] The maps are sampled by an equiangular sampling with bandwidth $b = 64$ ($n = 4b^2 = 16,384$ pixels) or an HEALPix sampling with $N_{side} = 32$ ($n = 12N_{side}^2 = 12,288$ pixels).

The equiangular graph is built with (4) and $k = 4$ neighbors (following Khasanova & Frossard, 2017). The HEALPix graph is built with (5), $k = 8$, and a kernel width $t$ set to the average of the distances (following Perraudin et al., 2019). The NN is made of 5 graph convolutional layers, each followed by a max pooling layer which down-samples by 4. A GAP and a fully connected layer with softmax follow. The polynomials are all of order $P = 3$ and the number of channels per layer is $16, 32, 64, 128, 256$, respectively. Following Esteves et al. (2018), the cross-entropy plus a triplet loss is optimized with Adam for 30 epochs on the dataset augmented by 3 random translations. The learning rate is $5 \cdot 10^{-2}$ and the batch size is 32.

Results are shown in table 1. As the network is trained for shape classification rather than retrieval, we report the classification F1 alongside the mAP used in the retrieval contest.[10] DeepSphere achieves the same performance as Cohen et al. (2018) and Esteves et al. (2018) at a much lower cost, suggesting that anisotropic filters are an unnecessary price to pay. As the information in those spherical maps resides in the low frequencies (figure 5), reducing the equivariance error didn't translate into improved performance. For the same reason, using the more uniform HEALPix sampling or lowering the resolution down to $N_{side} = 8$ ($n = 768$ pixels) didn't impact performance either.

---

[7]We however verified that the convolution is equivariant to longitudinal and latitudinal rotations, as intended.

[8]Albeit we didn't observe much improvement by using the convex hull.

[7]As implemented in `https://github.com/jonas-koehler/s2cnn`.

[10]We omit the F1 for Cohen et al. (2018) as we didn't get the mAP reported in the paper when running it.

|                                             | accuracy | time    |
|---------------------------------------------|----------|---------|
| Perraudin et al. (2019), 2D CNN baseline    | 54.2     | 104 ms  |
| Perraudin et al. (2019), CNN variant, $k = 8$ | 62.1   | 185 ms  |
| Perraudin et al. (2019), FCN variant, $k = 8$ | 83.8   | 185 ms  |
| $k = 8$ neighbors, $t$ from section 3.2     | 87.1     | 185 ms  |
| $k = 20$ neighbors, $t$ from section 3.2    | 91.3     | 250 ms  |
| $k = 40$ neighbors, $t$ from section 3.2    | 92.5     | 363 ms  |

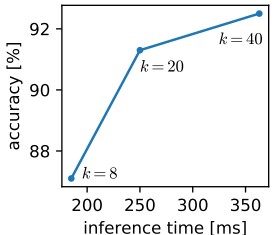

Table 2: Results on the classification of partial convergence maps. Lower equivariance error translates to higher performance.

Figure 7: Tradeoff between cost and accuracy.

## 4.2 COSMOLOGICAL MODEL CLASSIFICATION

Given observations, cosmologists estimate the posterior probability of cosmological parameters, such as the matter density $\Omega_m$ and the normalization of the matter power spectrum $\sigma_8$. Those parameters are typically estimated by likelihood-free inference, which requires a function to predict the parameters from simulations. As that is complicated to setup, prediction methods are typically benchmarked on the classification of spherical maps instead (Schmelzle et al., 2017). We used the same task, data, and setup as Perraudin et al. (2019): the classification of 720 partial convergence maps made of $n \approx 10^6$ pixels ($1/12 \approx 8\%$ of a sphere at $N_{side} = 1024$) from two $\Lambda$CDM cosmological models, ($\Omega_m = 0.31$, $\sigma_8 = 0.82$) and ($\Omega_m = 0.26$, $\sigma_8 = 0.91$), at a relative noise level of 3.5 (i.e., the signal is hidden in noise of 3.5 times higher standard deviation). Convergence maps represent the distribution of over- and under-densities of mass in the universe (see Bartelmann, 2010, for a review of gravitational lensing).

Graphs are built with (5), $k = 8, 20, 40$ neighbors, and the corresponding optimal kernel widths $t$ given in section 3.2. Following Perraudin et al. (2019), the NN is made of 5 graph convolutional layers, each followed by a max pooling layer which down-samples by 4. A GAP and a fully connected layer with softmax follow. The polynomials are all of order $P = 4$ and the number of channels per layer is $16, 32, 64, 64, 64$, respectively. The cross-entropy loss is optimized with Adam for 80 epochs. The learning rate is $2 \cdot 10^{-4} \cdot 0.999^{\text{step}}$ and the batch size is 8.

Unlike on SHREC'17, results (table 2) show that a lower equivariance error on the convolutions translates to higher performance. That is probably due to the high frequency content of those maps (figure 5). There is a clear cost-accuracy tradeoff, controlled by the number of neighbors $k$ (figure 7). This experiment moreover demonstrates DeepSphere's flexibility (using partial spherical maps) and scalability (competing spherical CNNs were tested on maps of at most $10,000$ pixels).

## 4.3 CLIMATE EVENT SEGMENTATION

We evaluate our method on a task proposed by (Mudigonda et al., 2017): the segmentation of extreme climate events, Tropical Cyclones (TC) and Atmospheric Rivers (AR), in global climate simulations (figure 1c). The data was produced by a 20-year run of the Community Atmospheric Model v5 (CAM5) and consists of 16 channels such as temperature, wind, humidity, and pressure at multiple altitudes. We used the pre-processed dataset from (Jiang et al., 2019).[11] There is 1,072,805 spherical maps, down-sampled to a level-5 icosahedral sampling ($n = 10 \cdot 4^l + 2 = 10,242$ pixels). The labels are heavily unbalanced with 0.1% TC, 2.2% AR, and 97.7% background (BG) pixels.

The graph is built with (5), $k = 6$ neighbors, and a kernel width $t$ set to the average of the distances. Following Jiang et al. (2019), the NN is an encoder-decoder with skip connections. Details in section C.3. The polynomials are all of order $P = 3$. The cross-entropy loss (weighted or non-weighted) is optimized with Adam for 30 epochs. The learning rate is $1 \cdot 10-3$ and the batch size is 64.

Results are shown in table 3 (details in tables 6, 7 and 8). The mean and standard deviation are computed over 5 runs. Note that while Jiang et al. (2019) and Cohen et al. (2019) use a weighted cross-entropy loss, that is a suboptimal proxy for the mAP metric. DeepSphere achieves state-of-

---

[11]Available at http://island.me.berkeley.edu/ugscnn/data.

|  | accuracy | mAP |
|---|---|---|
| Jiang et al. (2019) (rerun) | 94.95 | 38.41 |
| Cohen et al. (2019) (S2R) | 97.5 | 68.6 |
| Cohen et al. (2019) (R2R) | 97.7 | 75.9 |
| DeepSphere (weighted loss) | $97.8 \pm 0.3$ | $77.15 \pm 1.94$ |
| DeepSphere (non-weighted loss) | $87.8 \pm 0.5$ | $89.16 \pm 1.37$ |

Table 3: Results on climate event segmentation: mean accuracy (over TC, AR, BG) and mean average precision (over TC and AR). DeepSphere achieves state-of-the-art performance.

| order $P$ | temp. (from past temp.) | | | day (from temperature) | | | day (from precipitations) | | |
|---|---|---|---|---|---|---|---|---|---|
| | MSE | MAE | R2 | MSE | MAE | R2 | MSE | MAE | R2 |
| 0 | 10.88 | 2.42 | 0.896 | 0.10 | 0.10 | 0.882 | 0.58 | 0.42 | $-0.980$ |
| 4 | 8.20 | 2.11 | 0.919 | 0.05 | 0.05 | 0.969 | 0.50 | 0.18 | 0.597 |

Table 4: Prediction results on data from weather stations. Structure always improves performance.

the-art performance, suggesting again that anisotropic filters are unnecessary. Note that results from Mudigonda et al. (2017) cannot be directly compared as they don't use the same input channels.

Compared to Cohen et al. (2019)'s conclusion, it is surprising that S2R does worse than DeepSphere (which is limited to S2S). Potential explanations are (i) that their icosahedral projection introduces harmful distortions, or (ii) that a larger architecture can compensate for the lack of generality. We indeed observed that more feature maps and depth led to higher performance (section C.3).

### 4.4 UNEVEN SAMPLING

To demonstrate the flexibility of modeling the sampled sphere by a graph, we collected historical measurements from $n \approx 10,000$ weather stations scattered across the Earth.[12] The spherical data is heavily non-uniformly sampled, with a much higher density of weather stations over North America than the Pacific (figure 1d). For illustration, we devised two artificial tasks. A dense regression: predict the temperature on a given day knowing the temperature on the previous 5 days. A global regression: predict the day (represented as one period of a sine over the year) from temperature or precipitations. Predicting from temperature is much easier as it has a clear yearly pattern.

The graph is built with (5), $k = 5$ neighbors, and a kernel width $t$ set to the average of the distances. The equivariance property of the resulting graph has not been tested, and we don't expect it to be good due to the heavily non-uniform sampling. The NN is made of 3 graph convolutional layers. The polynomials are all of order $P = 0$ or $4$ and the number of channels per layer is $50, 100, 100$, respectively. For the global regression, a GAP and a fully connected layer follow. For the dense regression, a graph convolutional layer follows instead. The MSE loss is optimized with RMSprop for 250 epochs. The learning rate is $1 \cdot 10^{-3}$ and the batch size is 64.

Results are shown in table 4. While using a polynomial order $P = 0$ is like modeling each time series independently with an MLP, orders $P > 0$ integrate neighborhood information. Results show that using the structure induced by the spherical geometry always yields better performance.

## 5 CONCLUSION

This work showed that DeepSphere strikes an interesting, and we think currently optimal, balance between desiderata for a spherical CNN. A single parameter, the number of neighbors $k$ a pixel is connected to in the graph, controls the tradeoff between cost and equivariance (which is linked to performance). As computational cost and memory consumption scales linearly with the number of pixels, DeepSphere scales to spherical maps made of millions of pixels, a required resolution

---

[12]https://www.ncdc.noaa.gov/ghcn-daily-description

to faithfully represent cosmological and climate data. Also relevant in scientific applications is the flexibility offered by a graph representation (for partial coverage, missing data, and non-uniform samplings). Finally, the implementation of the graph convolution is straightforward, and the ubiquity of graph neural networks — pushing for their first-class support in DL frameworks — will make implementations even easier and more efficient.

A potential drawback of graph Laplacian-based approaches is the isotropy of graph filters, reducing in principle the expressive power of the NN. Experiments from Cohen et al. (2019) and Boscaini et al. (2016) indeed suggest that more general convolutions achieve better performance. Our experiments on 3D shapes (section 4.1) and climate (section 4.3) however show that DeepSphere's isotropic filters do not hinder performance. Possible explanations for this discrepancy are that NNs somehow compensate for the lack of anisotropic filters, or that some tasks can be solved with isotropic filters. The distortions induced by the icosahedral projection in (Cohen et al., 2019) or the leakage of curvature information in (Boscaini et al., 2016) might also alter performance.

Developing graph convolutions on irregular samplings that respect the geometry of the sphere is another research direction of importance. Practitioners currently interpolate their measurements (coming from arbitrarily positioned weather stations, satellites or telescopes) to regular samplings. This practice either results in a waste of resolution or computational and storage resources. Our ultimate goal is for practitioners to be able to work directly on their measurements, however distributed.

### ACKNOWLEDGMENTS

We thank Pierre Vandergheynst for advices, and Taco Cohen for his inputs on the intriguing results of our comparison with Cohen et al. (2019). We thank the anonymous reviewers for their constructive feedback. The following software packages were used for computation and plotting: PyGSP (Defferrard et al.), healpy (Zonca et al., 2019), matplotlib (Hunter, 2007), SciPy (Virtanen et al., 2020), NumPy (Walt et al., 2011), TensorFlow (Abadi et al., 2015).

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

# SUPPLEMENTARY MATERIAL

## A PROOF OF THEOREM 3.1

**Preliminaries.** The proof of theorem 3.1 is inspired from the work of Belkin & Niyogi (2008). As a result, we start by restating some of their results. Given a sampling $\mathcal{V} = \{x_i \in \mathcal{M}\}_{i=1}^n$ of a closed, compact and infinitely differentiable manifold $\mathcal{M}$, a smooth ($\in \mathcal{C}_\infty(\mathcal{M})$) function $f : \mathcal{M} \to \mathbb{R}$, and defined the vector $\boldsymbol{f}$ of samples of $f$ as follows: $T_\mathcal{V} f = \boldsymbol{f} \in \mathbb{R}^n$, $\boldsymbol{f}_i = f(x_i)$. The proof is constructed by leveraging 3 different operators:

- The extended graph Laplacian operator, already presented in (6), is a linear operator $L_n^t : L^2(\mathcal{M}) \to L^2(\mathcal{M})$ defined as

$$L_n^t f(y) := \frac{1}{n} \sum_{i=1}^n e^{-\frac{\|x_i - y\|^2}{4t}} \left( f(y) - f(x_i) \right). \tag{7}$$

  Note that we have the following relation $\boldsymbol{L}_n^t \boldsymbol{f} = T_\mathcal{V} L_n^t f$.
- The functional approximation to the Laplace-Beltrami operator is a linear operator $L^t : L^2(\mathcal{M}) \to L^2(\mathcal{M})$ defined as

$$L^t f(y) = \int_\mathcal{M} e^{-\frac{\|x - y\|^2}{4t}} \left( f(y) - f(x) \right) d\mu(x), \tag{8}$$

  where $\mu$ is the uniform probability measure on the manifold $\mathcal{M}$, and $\mathrm{vol}(\mathcal{M})$ is the volume of $\mathcal{M}$.
- The Laplace-Beltrami operator $\Delta_\mathcal{M}$ is defined as the divergence of the gradient

$$\Delta_\mathcal{M} f(y) := -\mathrm{div}(\nabla_\mathcal{M} f) \tag{9}$$

  of a differentiable function $f : \mathcal{M} \to \mathbb{R}$. The gradient $\nabla f : \mathcal{M} \to T_p\mathcal{M}$ is a vector field defined on the manifold pointing towards the direction of steepest ascent of $f$, where $T_p\mathcal{M}$ is the affine space of all vectors tangent to $\mathcal{M}$ at $p$.

Leveraging these three operators, Belkin & Niyogi (2008; 2007) have build proofs of both pointwise and spectral convergence of the extended graph Laplacian towards the Laplace-Beltrami operator in the general setting of any compact, closed and infinitely differentiable manifold $\mathcal{M}$, where the sampling $\mathcal{V}$ is drawn randomly on the manifold. For this reason, their results are all to be interpreted in a probabilistic sense. Their proofs consist in establishing that (6) converges in probability towards (8) as $n \to \infty$ and (8) converges towards (9) as $t \to 0$. In particular, this second step is given by the following:

**Proposition 1** (Belkin & Niyogi (2008), Proposition 4.4). *Let $\mathcal{M}$ be a $k$-dimensional compact smooth manifold embedded in some Euclidean space $\mathbb{R}^N$, and fix $y \in \mathcal{M}$. Let $f \in \mathcal{C}_\infty(\mathcal{M})$. Then*

$$\frac{1}{t} \frac{1}{(4\pi t)^{k/2}} L^t f(y) \xrightarrow{t \to 0} \frac{1}{vol(\mathcal{M})} \Delta_\mathcal{M} f(y). \tag{10}$$

**Building the proof.** As the sphere is a compact smooth manifold embedded in $\mathbb{R}^3$, we can reuse proposition 1. Thus, our strategy to prove Theorem 3.1 is to (i) show that

$$\lim_{n \to \infty} L_n^t f(y) = L^t(y) \tag{11}$$

for a particular class of *deterministic* samplings, and (ii) apply Proposition 1.

We start by proving that for smooth functions, for any fixed $t$, the extended graph Laplacian $L_n^t$ converges towards its continuous counterpart $L^t$ as the sampling increases in size.

**Proposition 2.** *For an equal area sampling $\{x_i \in \mathbb{S}^2\}_{i=1}^n : A_i = A_j \forall i, j$ of the sphere it is true that for all $f : \mathbb{S}^2 \to \mathbb{R}$ Lipschitz with respect to the Euclidean distance $\|\cdot\|$ with Lipschitz constant $C_f$*

$$\left| \int_{\mathbb{S}^2} f(x) d\mu(x) - \frac{1}{n} \sum_i f(x_i) \right| \leq C_f d^{(n)}.$$

*Furthermore, for all $y \in \mathbb{S}^2$ the Heat Kernel Graph Laplacian operator $L_n^t$ converges pointwise to the functional approximation of the Laplace Beltrami operator $L^t$*

$$L_n^t f(y) \xrightarrow{n \to \infty} L^t f(y).$$

*Proof.* Assuming $f : \mathbb{S}^2 \to \mathbb{R}$ is Lipschitz with Lipschitz constant $C_f$, we have

$$\left| \int_{\sigma_i} f(x) \mathrm{d}\mu(x) - \frac{1}{n} f(x_i) \right| \leq C_f d^{(n)} \frac{1}{n},$$

where $\sigma_i \subset \mathbb{S}^2$ is the subset of the sphere corresponding to the patch around $x_i$. Remember that the sampling is equal area. Hence, using the triangular inequality and summing all the contributions of the $n$ patches, we obtain

$$\left| \int_{\mathbb{S}^2} f(x) \mathrm{d}\mu(x) - \frac{1}{n} \sum_i f(x_i) \right| \leq \sum_i \left| \frac{1}{4\pi^2} \int_{\sigma_i} f(x) \mathrm{d}\mu(x) - \frac{1}{n} f(x_i) \right| \leq n C_f d^{(n)} \frac{1}{n} = C_f d^{(n)}$$

A direct application of this result leads to the following pointwise convergences

$$\forall f \text{ Lipschitz}, \quad \forall y \in \mathbb{S}^2, \qquad \frac{1}{n} \sum_i e^{-\frac{\|x_i - y\|^2}{4t}} \to \int e^{-\frac{\|x - y\|^2}{4t}} d\mu(x)$$

$$\forall f \text{ Lipschitz}, \quad \forall y \in \mathbb{S}^2, \qquad \frac{1}{n} \sum_i e^{-\frac{\|x_i - y\|^2}{4t}} f(x_i) \to \int e^{-\frac{\|x - y\|^2}{4t}} f(x) d\mu(x)$$

Definitions 6 and 8 end the proof. $\qquad \square$

The last proposition show that for a *fixed* $t$, $L_n^t f(x) \to 1/4\pi^2 L^t f(x)$. To utilize Proposition 1 and complete the proof, we need to find a sequence of $t_n$ for which this holds as $t_n \to 0$. Furthermore this should hold with a faster decay than $\frac{1}{4\pi t_n^2}$.

**Proposition 3.** *Given a sampling regular enough, i.e., for which we assume $A_i = A_j \ \forall i, j$ and $d^{(n)} \leq \frac{C}{n^\alpha}$, $\alpha \in (0, 1/2]$, a Lipschitz function $f$ and a point $y \in \mathbb{S}^2$ there exists a sequence $t_n = n^\beta, \beta < 0$ such that*

$$\forall f \text{ Lipschitz}, \forall x \in \mathbb{S}^2 \quad \left| \frac{1}{4\pi t_n^2} \left( L_n^{t_n} f(x) - L^{t_n} f(x) \right) \right| \xrightarrow{n \to \infty} 0.$$

*Proof.* To ease the notation, we define

$$K^t(x, y) := e^{-\frac{\|x - y\|^2}{4t}} \tag{12}$$

$$\phi^t(x; y) := e^{-\frac{\|x - y\|^2}{4t}} \left( f(y) - f(x) \right). \tag{13}$$

We start with the following inequality

$$\|L_n^t f - L^t f\|_\infty = \max_{y \in \mathbb{S}^2} \left| L_n^t f(y) - L^t f(y) \right|$$

$$= \max_{y \in \mathbb{S}^2} \left| \frac{1}{n} \sum_{i=1}^n \phi^t(x_i; y) - \int_{\mathbb{S}^2} \phi^t(x; y) d\mu(x) \right|$$

$$\leq \max_{y \in \mathbb{S}^2} \sum_{i=1}^n \left| \frac{1}{n} \phi^t(x_i; y) - \int_{\sigma_i} \phi^t(x; y) d\mu(x) \right|$$

$$\leq d^{(n)} \max_{y \in \mathbb{S}^2} C_{\phi_y^t}, \tag{14}$$

where $C_{\phi_y^t}$ is the Lipschitz constant of $x \to \phi^t(x, y)$ and the last inequality follows from Proposition 2. Using the assumption $d^{(n)} \leq \frac{C}{\sqrt{n}}$ we find

$$\|L_n^t f - L^t f\|_\infty \leq \frac{C}{\sqrt{n}} \max_{y \in \mathbb{S}^2} C_{\phi_y^t}$$

We now find the explicit dependence between $t$ and $C_{\phi_y^t}$

$$
\begin{aligned}
C_{\phi_y^t} &= \|\partial_x \phi^t(\cdot; y)\|_\infty \\
&= \|\partial_x \left(K^t(\cdot; y)f\right)\|_\infty \\
&= \|\partial_x K^t(\cdot; y)f + K^t(\cdot; y)\partial_x f\|_\infty \\
&\leq \|\partial_x K^t(\cdot; y)f\|_\infty + \|K^t(\cdot; y)\partial_x f\|_\infty \\
&\leq \|\partial_x K^t(\cdot; y)\|_\infty \|f\|_\infty + \|K^t(\cdot; y)\|_\infty \|\partial_x f\|_\infty \\
&= \|\partial_x K^t(\cdot; y)\|_\infty \|f\|_\infty + \|\partial_x f\|_\infty \\
&= C_{K_y^t}\|f\|_\infty + \|\partial_x f\|_\infty \\
&= C_{K_y^t}\|f\|_\infty + C_f
\end{aligned}
$$

where $C_{K_y^t}$ is the Lipschitz constant of the function $x \to K^t(x; y)$. We note that this constant does not depend on $y$:

$$
C_{K_y^t} = \left\|\partial_x e^{-\frac{x^2}{4t}}\right\|_\infty = \left\|\frac{x}{2t}e^{-\frac{x^2}{4t}}\right\|_\infty = \frac{x}{2t}e^{-\frac{x^2}{4t}}\Big|_{x=\sqrt{2t}} = (2et)^{-\frac{1}{2}} \propto t^{-\frac{1}{2}}.
$$

Hence we have

$$
\frac{C}{\sqrt{n}} \max_{y \in \mathbb{S}^2} C_{\phi_y^t} \leq \frac{C}{\sqrt{n}} \left((2et)^{-\frac{1}{2}} \|f\|_\infty + C_f\right)
$$

$$
\leq \frac{C\|f\|_\infty}{n^\alpha(2et)^{1/2}} + \frac{C}{n^\alpha}C_f.
$$

Inculding this result in (14) and rescaling by $1/4\pi t^2$, we obtain

$$
\left\|\frac{1}{4\pi t^2}\left(L_n^t f - L^t f\right)\right\|_\infty \leq \frac{1}{4\pi t^2}\left\|\left(L_n^t f - L^t f\right)\right\|_\infty
$$

$$
\leq \frac{C}{4\pi}\left[\frac{\|f\|_\infty}{\sqrt{2e}}\frac{1}{n^\alpha t^{5/2}} + \frac{C_f}{n^\alpha t^2}\right].
$$

In order for $\frac{C}{4\pi}\left[\frac{\|f\|_\infty}{\sqrt{2e}}\frac{1}{n^\alpha t^{5/2}} + \frac{C_f}{n^\alpha t^2}\right] \xrightarrow[t\to 0]{n\to\infty} 0$, we need $\begin{cases} n^\alpha t^{5/2} \to \infty \\ n^\alpha t^2 \to \infty \end{cases}$

It happens if $\begin{cases} t(n) = n^\beta, & \beta \in \left(-\frac{2\alpha}{5}, 0\right) \\ t(n) = n^\beta, & \beta \in \left(-\frac{\alpha}{2}, 0\right) \end{cases} \implies t(n) = n^\beta, \quad \beta \in \left(-\frac{2\alpha}{5}, 0\right).$

Indeed, we have

$n^\alpha t^{5/2} = n^{5/2\beta+\alpha} \xrightarrow{n\to\infty} \infty$ since $\frac{5}{2}\beta + \alpha > 0 \iff \beta > -\frac{2\alpha}{5}$

and $n^\alpha t^2 = n^{2\beta+\alpha} \xrightarrow{n\to\infty} \infty$ since $2\beta + \alpha > 0 \iff \beta > -\frac{\alpha}{2}$.

As a result, for $t = n^\beta$ with $\beta \in \left(-\frac{1}{5}, 0\right)$ we have $\begin{cases} (t_n) \xrightarrow{n\to\infty} 0 \\ \left\|\frac{1}{4\pi t_n^2}L_n^{t_n}f - \frac{1}{4\pi t_n^2}L^{t_n}f\right\|_\infty \xrightarrow{n\to\infty} 0, \end{cases}$

which concludes the proof. $\qquad\square$

Theorem 3.1, is then an immediate consequence of Proposition 3 and 1.

*Proof of Theorem 3.1.* Thanks to Proposition 3 and Proposition 1 we conclude that $\forall y \in \mathbb{S}^2$

$$
\lim_{n\to\infty} \frac{1}{4\pi t_n^2}L_n^{t_n}f(y) = \lim_{n\to\infty} \frac{1}{4\pi t_n^2}L^{t_n}f(y) = \frac{1}{|\mathbb{S}^2|}\Delta_{\mathbb{S}^2}f(y)
$$

$\qquad\square$

In (Belkin & Niyogi, 2008), the sampling is drawn from a uniform random distribution on the sphere, and their proof heavily relies on the uniformity properties of the distribution from which the sampling is drawn. In our case the sampling is deterministic, and this is indeed a problem that we need to overcome by imposing the regularity conditions above.

| | micro (label average) | | | | macro (instance average) | | | |
|---|---|---|---|---|---|---|---|---|
| | P@N | R@N | F1@N | mAP | P@N | R@N | F1@N | mAP |
| Cohen et al. (2018) ($b = 128$) | 0.701 | 0.711 | 0.699 | 0.676 | - | - | - | - |
| Cohen et al. (2018) (simplified, $b = 64$) | 0.704 | 0.701 | 0.696 | 0.665 | 0.430 | 0.480 | 0.429 | 0.385 |
| Esteves et al. (2018) ($b = 64$) | 0.717 | 0.737 | - | 0.685 | 0.450 | 0.550 | - | 0.444 |
| DeepSphere (equiangular $b = 64$) | 0.709 | 0.700 | 0.698 | 0.665 | 0.439 | 0.489 | 0.439 | 0.403 |
| DeepSphere (HEALPix $N_{side} = 32$) | 0.725 | 0.717 | 0.715 | 0.686 | 0.475 | 0.508 | 0.468 | 0.428 |

Table 5: Official metrics from the SHREC'17 object retrieval competition.

To conclude, we see that the result obtained is of similar form than the result obtained in (Belkin & Niyogi, 2008). Given the kernel density $t(n) = n^{\beta}$, Belkin & Niyogi (2008) proved convergence in the random case for $\beta \in (-\frac{1}{4}, 0)$ and we proved convergence in the deterministic case for $\beta \in (-\frac{2\alpha}{5}, 0)$, where $\alpha \in (0, 1/2]$ (for the spherical manifold).

## B  Proof of Theorem 3.2

*Proof.* Fix $x \in \mathbb{S}^2$. Since any rotation $R(g)$ is an isometry, and the Laplacian $\Delta$ commutes with all isometries of a Riemanniann manifold, and defining $R(g)f =: f'$ for ease of notation, we can write that

$$\left| R(g)\hat{L}_n^{t_n} f(x) - \hat{L}_n^{t_n} R(g)f(x) \right| \le \left| R(g)\hat{L}_n^{t_n} f(x) - R(g)\Delta_{\mathbb{S}^2} f(x) \right| + \left| R(g)\Delta_{\mathbb{S}^2} f(x) - \hat{L}_n^{t_n} R(g)f(x) \right| =$$

$$= \left| R(g)(\hat{L}_n^{t_n} f - \Delta_{\mathbb{S}^2} f)(x) \right| + \left| \Delta_{\mathbb{S}^2} f'(x) - \hat{L}_n^{t_n} f'(x) \right| \le$$

$$\le \left| (\hat{L}_n^{t_n} f - \Delta_{\mathbb{S}^2} f)(g^{-1}(x)) \right| + \left| \Delta_{\mathbb{S}^2} f'(x) - \hat{L}_n^{t_n} f'(x) \right|$$

Since $g^{-1}(x) \in \mathbb{S}^2$ and $f'$ still satisfies hypothesis, we can apply theorem 3.1 to say that

$$\left| (\hat{L}_n^{t_n} f - \Delta_{\mathbb{S}^2} f)(g^{-1}(x)) \right| \xrightarrow{n\to\infty} 0$$

$$\left| \Delta_{\mathbb{S}^2} f'(x) - \hat{L}_n^{t_n} f'(x) \right| \xrightarrow{n\to\infty} 0$$

to conclude that

$$\forall x \in \mathbb{S}^2 \quad \left| R(g)\hat{L}_n^{t_n} f(x) - \hat{L}_n^{t_n} R(g)f(x) \right| \xrightarrow{n\to\infty} 0$$

$\square$

## C  Experimental details

### C.1  3D objects recognition

Table 5 shows the results obtained from the SHREC'17 competition's official evaluation script.

$$[GC_{16} + BN + ReLU]_{nside32} + \text{Pool} + [GC_{32} + BN + ReLU]_{nside16} + \text{Pool}$$
$$+ [GC_{64} + BN + ReLU]_{nside8} + \text{Pool} + [GC_{128} + BN + ReLU]_{nside4} \quad (15)$$
$$+ \text{Pool} + [GC_{256} + BN + ReLU]_{nside2} + \text{Pool} + GAP + FCN + \text{softmax}$$

### C.2  Cosmological model classification

$$[GC_{16} + BN + ReLU]_{nside1024} + \text{Pool} + [GC_{32} + BN + ReLU]_{nside512}$$
$$+ \text{Pool} + [GC_{64} + BN + ReLU]_{nside256} + \text{Pool} \quad (16)$$
$$+ [GC_{64} + BN + ReLU]_{nside128} + \text{Pool} + [GC_{64} + BN + ReLU]_{nside64}$$
$$+ \text{Pool} + [GC_2]_{nside32} + GAP + \text{softmax}$$

|  | TC | AR | BG | mean |
|---|---|---|---|---|
| Mudigonda et al. (2017) | 74 | 65 | 97 | 78.67 |
| Jiang et al. (2019) (paper) | 94 | 93 | 97 | 94.67 |
| Jiang et al. (2019) (rerun) | 93.9 | 95.7 | 95.2 | 94.95 |
| Cohen et al. (2019) (S2R) | 97.8 | 97.3 | 97.3 | 97.5 |
| Cohen et al. (2019) (R2R) | 97.9 | 97.8 | 97.4 | 97.7 |
| DS (Jiang architecture, weighted loss) | 97.1 | 97.6 | 96.5 | 97.1 |
| DS (weighted loss) | $97.4 \pm 1.1$ | $97.7 \pm 0.7$ | $98.2 \pm 0.5$ | $97.8 \pm 0.3$ |
| DS (wider architecture, weighted loss) | 91.5 | 93.4 | 99.0 | 94.6 |
| DS (Jiang architecture, non-weighted loss) | 33.6 | 93.6 | 99.3 | 75.5 |
| DS (non-weighted loss) | $69.2 \pm 3.7$ | $94.5 \pm 2.9$ | $99.7 \pm 0.1$ | $87.8 \pm 0.5$ |
| DS (wider architecture, non-weighted loss) | 73.4 | 92.7 | 99.8 | 88.7 |

Table 6: Results on climate event segmentation: accuracy. Tropical cyclones (TC) and atmospheric rivers (AR) are the two positive classes, against the background (BG). Mudigonda et al. (2017) is not directly comparable as they don't use the same input feature maps. Note that a non-weighted cross-entropy loss is not optimal for the accuracy metric.

|  | TC | AR | mean |
|---|---|---|---|
| Jiang et al. (2019) (rerun) | 11.08 | 65.21 | 38.41 |
| Cohen et al. (2019) (S2R) | - | - | 68.6 |
| Cohen et al. (2019) (R2R) | - | - | 75.9 |
| DS (Jiang architecture, non-weighted loss) | 46.2 | 93.9 | 70.0 |
| DS (non-weighted loss) | $80.86 \pm 2.42$ | $97.45 \pm 0.38$ | $89.16 \pm 1.37$ |
| DS (wider architecture, non-weighted loss) | 84.71 | 98.05 | 91.38 |
| DS (Jiang architecture, weighted loss) | 49.7 | 89.2 | 69.5 |
| DS (weighted loss) | $58.88 \pm 3.17$ | $95.41 \pm 1.51$ | $77.15 \pm 1.94$ |
| DS (wider architecture, weighted loss) | 52.80 | 94.78 | 73.79 |

Table 7: Results on climate event segmentation: average precision. Tropical cyclones (TC) and atmospheric rivers (AR) are the two positive classes. Note that a weighted cross-entropy loss is not optimal for the average precision metric.

## C.3 CLIMATE EVENT SEGMENTATION

Table 6, 7, and 8 show the accuracy, mAP, and efficiency of all the NNs we ran.

The experiment with the model from Jiang et al. (2019) was rerun in order to obtain the AP metrics, but with a batch size of 64 instead of 256 due to GPU memory limit.

Several experiments were run with different architectures for DeepSphere (DS). Jiang architecture use a similar one as Jiang et al. (2019), with only the convolutional operators replaced. DeepSphere only is the original architecture giving the best results, deeper and with four times more feature maps than Jiang architecture. And the wider architecture is the same as the previous one with two times the number of feature maps.

Regarding the weighted loss, the weights are chosen with `scikit-learn` function `compute_class_weight` on the training set.

| | size | speed | |
|---|---|---|---|
| | params | inference | training |
| Jiang et al. (2019) | 330 k | 10 ms | 10 h |
| DeepSphere (Jiang architecture) | 590 k | 5 ms | 3 h |
| DeepSphere | 13 M | 33 ms | 13 h |
| DeepSphere (wider architecture) | 52 M | 50 ms | 20 h |

Table 8: Results on climate event segmentation: size and speed.

DeepSphere with Jiang architecture
encoder:

$$[GC_8 + BN + ReLU]_{L5} + \text{Pool} + [GC_{16} + BN + ReLU]_{L4} + \text{Pool}$$
$$+ [GC_{32} + BN + ReLU]_{L3} + \text{Pool} + [GC_{64} + BN + ReLU]_{L2} + \text{Pool} \tag{17}$$
$$+ [GC_{128} + BN + ReLU]_{L1} + \text{Pool} + [GC_{128} + BN + ReLU]_{L0}$$

decoder:

$$\text{Unpool} + [GC_{128} + BN + ReLU]_{L1} + \text{concat} + [GC_{128} + BN + ReLU]_{L1}$$
$$+ \text{Unpool} + [GC_{64} + BN + ReLU]_{L2} + \text{concat}$$
$$+ [GC_{64} + BN + ReLU]_{L2} + \text{Unpool} + [GC_{32} + BN + ReLU]_{L3} \tag{18}$$
$$+ \text{concat} + [GC_{32} + BN + ReLU]_{L3} + \text{Unpool}$$
$$+ [GC_{16} + BN + ReLU]_{L4} + \text{concat} + [GC_{16} + BN + ReLU]_{L4} + \text{Unpool}$$
$$+ [GC_8 + BN + ReLU]_{L5} + \text{concat} + [GC_8 + BN + ReLU]_{L5} + [GC_3]_{L5}$$

Concat is the operation that concatenate the results of the corresponding encoder layer.

Original DeepSphere architecture with encoder decoder
encoder:

$$[GC_{32} + BN + ReLU]_{L5} + [GC_{64} + BN + ReLU]_{L5}$$
$$+ \text{Pool} + [GC_{128} + BN + ReLU]_{L4} + \text{Pool} \tag{19}$$
$$+ [GC_{256} + BN + ReLU]_{L3} + \text{Pool} + [GC_{512} + BN + ReLU]_{L2}$$
$$+ \text{Pool} + [GC_{512} + BN + ReLU]_{L1} + \text{Pool} + [GC_{512}]_{L0}$$

decoder:

$$\text{Unpool} + [GC_{512} + BN + ReLU]_{L1} + \text{concat} + [GC_{512} + BN + ReLU]_{L1}$$
$$+ \text{Unpool} + [GC_{256} + BN + ReLU]_{L2} + \text{concat}$$
$$+ [GC_{256} + BN + ReLU]_{L2} + \text{Unpool} + [GC_{128} + BN + ReLU]_{L3} \tag{20}$$
$$+ \text{concat} + [GC_{128} + BN + ReLU]_{L3} + \text{Unpool}$$
$$+ [GC_{64} + BN + ReLU]_{L4} + \text{concat} + [GC_{64} + BN + ReLU]_{L4}$$
$$+ \text{Unpool} + [GC_{32} + BN + ReLU]_{L5} + [GC_3]_{L5}$$

## C.4 UNEVEN SAMPLING

Architecture for dense regression:

$$[GC_{50} + BN + ReLU] + [GC_{100} + BN + ReLU] + [GC_{100} + BN + ReLU] + [GC_1] \quad (21)$$

Architecture for global regression:

$$
\begin{aligned}
&[GC_{50} + BN + ReLU] + [GC_{100} + BN + ReLU] \\
&+ [GC_{100} + BN + ReLU] + GAP + FCN
\end{aligned} \quad (22)
$$

