# OpenReview forum: "DeepSphere: a graph-based spherical CNN"
_ICLR.cc/2020/Conference — Accept (Spotlight)_

### Official Review · AnonReviewer1 · 2019-10-22
**Official Blind Review #1**

**Rating:** 6

**Review:**

In this paper, CNNs specialized for spherical data are studied. The proposed architecture is a combination of existing frameworks based on the discretization of a sphere as a graph. As a main result, the paper shows a convergence result, which is related to the rotation equivalence on a sphere. The experiments show the proposed model achieves a good tradeoff between the prediction performance and the computational cost.

Although the theoretical result is not strong enough, the empirical results show the proposed approach is promising. Therefore I vote for acceptance.

The paper is overall clearly written. It is nice that the authors try to mitigate from overclaiming of the analysis.

As a non-expert of spherical CNN, I don't understand clearly the gap between the result Theorem 3.1 and showing the rotation equivalence. It would be nice to add some counterexample (i.e., in what situation the proposed approach does not have rotational equivalence while Theorem 3.1 holds).

**Experience Assessment:**

I do not know much about this area.

**Review Assessment: Checking Correctness Of Derivations And Theory:**

I assessed the sensibility of the derivations and theory.

**Review Assessment: Checking Correctness Of Experiments:**

I did not assess the experiments.

**Review Assessment: Thoroughness In Paper Reading:**

I made a quick assessment of this paper.

---

> ### Author Response · Authors · 2019-11-11
> **Answer to Official Blind Review #1**
>
> We thank the reviewer for their time assessing our work and their constructive feedback.
>
> We prepared a revised manuscript, to be uploaded shortly, containing a deeper theoretical discussion closing the gap between Theorem 3.1 and rotation equivariance. In a small proposition, to be added after theorem 3.1, we precise mathematically the relationship between these two concepts. In short, we proved that, if theorem 3.1 holds, our graph Laplacian L commutes with any rotation operator R in the limit of infinite sampling (pointwise), i.e., |LRf(x) - RLf(x)| -> 0, thus answering the reviewer's concerns about this subject.

---

### Official Review · AnonReviewer3 · 2019-10-23
**Official Blind Review #3**

**Rating:** 6

**Review:**

The paper studies the problem of designing a convolution for a spherical neural network. The authors use the existing graph CNN formulation and a pooling strategy that exploits hierarchical pixelations of the sphere to learn from the discretized sphere. The main idea is to model the discretized sphere as a graph of connected pixels: the length of the shortest path between two pixels is an approximation of the geodesic distance between them. To show the computational efficiency, sampling flexibility and rotation equivariance, extensive experiments are conducted, including 3D object recognition, cosmological mode classification, climate event segmentation and uneven sampling.
Pros:
1. The application and combination of different techniques in this paper are smart.
2. The experiments show that the proposed method outperforms other baseline methods.
3. The paper is well organized and written.
Cons:
1. It is a good application of known techniques, but the novelty is limited.
2. It is suggested to add more baselines in the experiments.

[1] Michael Defferrard, Xavier Bresson, and Pierre Vandergheynst. Convolutional neural networks on graphs with fast localized spectral filtering. In Advances in Neural Information ProcessingSystems, 2016

**Experience Assessment:**

I have published one or two papers in this area.

**Review Assessment: Checking Correctness Of Derivations And Theory:**

I carefully checked the derivations and theory.

**Review Assessment: Checking Correctness Of Experiments:**

I carefully checked the experiments.

**Review Assessment: Thoroughness In Paper Reading:**

I read the paper thoroughly.

---

> ### Author Response · Authors · 2019-11-11
> **Answer to Official Blind Review #3**
>
> We thank the reviewer for their time assessing our work and their constructive feedback.
>
> While novelty might be limited (although we'd argue that designing a good graph is non-trivial, if only by checking how many papers have been written on the convergence / consistency of discrete Laplacians), potential impact is certainly not. Researchers working with large spherical maps, in multiple fields, will benefit from the possibility to tackle their problems with a neural network.
>
> Which other baselines would you like to see? We compared with previous works that tackled the same tasks. It is difficult (and probably unfair) to adapt baselines not designed to solve those tasks.

---

### Official Review · AnonReviewer2 · 2019-10-24
**Official Blind Review #2**

**Rating:** 8

**Review:**


The paper presents DeepSphere, a method for learning over spherical data via a graphical representation and graph-convolutions. The primary goal is to develop a method that encodes equivariance to rotations, cheaply. The graph is formed by sampling the surface of the sphere and connecting neighbors according to a distance-based similarity measure. The equivariance of the representation is demonstrated empirically and theoretical background on its convergence properties are shown. DeepSphere is then demonstrated on several problems as well as shown how it applies to non-uniform data.

The paper is interesting and clear. The projection of structured data to graphical representations is both efficient in utilizing existing algorithmic techniques for graph convolutions and useful for approaching the spherical structure of the data. The theoretical analysis and discussion of sampling is interesting, though should be more clearly stated throughout and potentially visualized in figures.

The experiments performed are thorough and interesting. The approach both outperforms baselines in inference time and accuracy. However, one wonders the performance on the well-researched tasks such as the performance on 3D imagery, e.g., Su & Grauman, 2017; Coors et al., 2018.

The unevenly sampled data is a nice extension showing the generality of the approach. How does the approach work for data connected within a radius rather than a k-nearest approach?

Minor:
- A figure detailing the parameters and setup for theorem 3.1 and figure 2 would be useful.
- The statement on the dispersion of the sampling sequence states “the smallest ball in \R^3 containing \sigma_i”, but I believe it should be “containing only \sigma_i”.

**Experience Assessment:**

I have read many papers in this area.

**Review Assessment: Checking Correctness Of Derivations And Theory:**

I assessed the sensibility of the derivations and theory.

**Review Assessment: Checking Correctness Of Experiments:**

I assessed the sensibility of the experiments.

**Review Assessment: Thoroughness In Paper Reading:**

I read the paper at least twice and used my best judgement in assessing the paper.

---

> ### Author Response · Authors · 2019-11-11
> **Answer to Official Blind Review #2**
>
> We thank the reviewer for their time assessing our work and their constructive feedback.
>
> We deliberately excluded experiments on omnidirectional imagery. In our opinion, those don't possess full spherical symmetries as gravity is orienting the objects. We encourage the reviewer to check the work of Khasanova and Frossard, who explicitly designed graph-based spherical CNNs for omnidirectional imaging. In [1], they designed a graph that yields an equivariant convolution to a subgroup of SO(3). Longitudinal rotations are equivariant by construction of the equiangular sampling, and they optimized the graph for latitudinal equivariance. Their scheme is presented in section 3.2 of our paper. While their convolution is not equivariant to the whole of SO(3), that is not an issue for this application as gravity prevents objects from rotating around the third axis. It may even be beneficial. Moreover, the orientation given by gravity allows to factorize the spherical graph and design anisotropic filters [2].
>
> Radius or kNN graphs are means to get a sparse graph for O(n) matrix multiplication, instead of O(n²) for the full distance-based similarity graph. We believe that the choice of one or the other doesn't really matter. Sparsification can be seen as a numerical approximation that replaces small values by zeros. The kNN scheme is often preferred in practice as the choice of k is directly linked to the computational cost, while the choice of a radius large enough to avoid disconnected vertices might include many more edges than necessary on denser areas.
>
> Thanks for pointing out an unclear statement about the dispersion of the sampling sequence. d_i should be understood as the largest distance between the center x_i and any point on the surface sigma_i. Hence, we define d_i to be the radius of the smallest ball centered at x_i containing sigma_i. We'll clarify.
>
> From the following two sentences, we don't understand what could be improved.
> * "The theoretical analysis and discussion of sampling is interesting, though should be more clearly stated throughout and potentially visualized in figures."
> * "A figure detailing the parameters and setup for theorem 3.1 and figure 2 would be useful."
> We would be glad if the reviewer could elaborate.
>
> A revised manuscript will be uploaded shortly.
>
> [1] Renata Khasanova and Pascal Frossard. Graph-based classification of omnidirectional images. In Proceedings of the IEEE International Conference on Computer Vision, 2017.
> [2] Renata Khasanova and Pascal Frossard. Geometry Aware Convolutional Filters for Omnidirectional Images Representation. In International Conference on Machine Learning. 2019.

---

### Public Comment · ~Jialin_Liu3 · 2019-11-07
**Two Questions about Rotation Equivariance**

Really interesting work.
I've got 2 questions:
1, As for efficiency, why not use GCN proposed by T.K. Kipf, the successor of ChebNet?
2. As for rotation equivariance, CCN states that "It is, however, possible to construct a CNN in which the activations transform in a predictable and reversible way," I understand what is reversible(invertible) in this work is what CCN calls activation, what is reversible in CCN is the rotation operator in this work, are they same?

Thanks.

Ref.
1. Semi-Supervised Classification with Graph Convolutional Networks. ICLR'17.
2. Covariant Compositional Networks For Learning Graphs. ICLR'18

---

### Author Response · Authors · 2019-11-15
**New revision**

We uploaded an improved manuscript thanks to the reviewers' comments.

The main update is the addition of theorem 3.2 that formalizes the relation between theorem 3.1 and rotation equivariance. Small changes across the text have been made to clarify the exposition further.

A link to a public git repository containing all the code will be added after the blind-review process.

---

### Decision · Program_Chairs · 2019-12-19

**Decision:**

Accept (Spotlight)

**Comment:**

This paper proposes a novel methodology for applying convolutional networks to spherical data through a graph-based discretization.   The reviewers all found the methodology sensible and the experiments convincing.  A common concern of the reviewers was the amount of novelty in the approach, as in it involves the combination of established methods, but ultimately they found that the empirical performance compared to baselines outweighed this.